# Photodynamic Therapy in Endodontics: A Helpful Tool to Combat Antibiotic Resistance? A Literature Review

**DOI:** 10.3390/antibiotics10091106

**Published:** 2021-09-13

**Authors:** Haitham Abdelkarim-Elafifi, Isabel Parada-Avendaño, Josep Arnabat-Dominguez

**Affiliations:** 1Faculty of Medicine and Health Sciences, University of Barcelona, 08907 Barcelona, Spain; haitham1136@gmail.com; 2Analysis and Design in Clinical Investigation, University of Barcelona, 08017 Barcelona, Spain; cotihums@gmail.com; 3Idibell Institute, 08908 Barcelona, Spain

**Keywords:** photodynamic therapy, antimicrobial photodynamic therapy, antibiotic resistance, endodontics, dentistry

## Abstract

Background: Antibiotic resistance has become a growing global problem where overprescription is a contributing factor for its development. In the endodontics field, complementary treatments, such as antimicrobial photodynamic therapy (aPDT), have been described to eliminate residual bacteria from the root canal space and reduce complications. The aim of this review is to describe the literature evidence up to now regarding the advantages, efficiency, and clinical outcomes of this therapy in endodontics as a possible tool to combat antibiotic resistance. Methods: A review of the literature from 2010 to 2021 was carried out using the PubMed and Web of Science databases. Two steps were taken: First, articles were compiled through the terms and MeSH terms “Photochesdmotherapy” and “endodontics.” Then, a second search was conducted using “photodynamic therapy” and “antibiotic resistance” or “drug resistance, microbial.” Results: A total of 51 articles were included for evaluation: 27 laboratory studies, 14 reviews, and 10 clinical studies. Laboratory studies show that aPDT achieves significant bacterial elimination, even against antibiotic-resistant species, and is also effective in biofilm disruption. Clinical studies suggest that aPDT can be considered a promising technique to reduce bacterial complications, and reviews about the issue confirm its advantages. Conclusion: The benefits of aPDT in reducing complications due to its antibacterial effects means a possible decrease in systemic antibiotic prescription in endodontics. In addition, it could be an alternative to local intracanal antibiotic therapy, avoiding the appearance of possible antibiotic resistance, as no bacterial resistance with aPDT has been described to date.

## 1. Introduction

The challenge of a successful outcome in root canal treatment, apart from the anatomical variations of the root canal systems, the tooth position, presence of calcified canals or pulp stones, and patient related factors, is the variety of bacterial strains that can be found in endodontic infections, such as *Streptococci*, *Peptostreptococcus*, *Lactobacilli*, *Propionibacterium*, *Actinomyces*, *Eubacterium*, *Veillonella parvula*, *Bacteroides*, *and Fusobacterium*, among others [1]. In addition, microbial contamination of the root canal system is not limited to the pulp tissue space, but can also penetrate the dentinal tubules up to a depth of 1000 µm, as well as accessory canals, anastomoses, and the apical complex anatomy. This makes adequate decontamination difficult with the classical treatment based on chemomechanical disinfection (CMD) together with irrigation solutions, with sodium hypochlorite (NaOCl) being the irrigant of choice during the procedure [2,3,4].

Following the literature evidence, the use of antibiotics in conjunction with the endodontic treatment is indicated in cases of an acute apical abscess when the patient has underlying medical complications or if the patient is healthy, but the infection shows signs of systemic involvement such as fever or swelling. Antibiotics are also indicated in cases of infections with a rapid onset (acute phase development in less than 24 h) and in spreading infections that cause cellulitis or osteomyelitis. In addition, they can be administered in persistent infections when all local measures and intracanal medications fail to resolve the infection. On the contrary, dentists should not prescribe antibiotics for symptomatic irreversible pulpitis, pulp necrosis, acute apical periodontitis, or chronic apical abscess [5].

Antibiotic administration is usually empirical, based on epidemiological data. Broad-spectrum amoxicillin is the most commonly used, although other practitioners prefer the combination of amoxicillin and clavulanic acid because of the low bacterial resistance to this combination [6,7,8].

Prescription by dentists represents 10% of all antibiotics prescribed by health care professionals [9,10], and their use is increasing, sometimes in the absence of indication, according to the current scientific evidence. This contributes to the serious public health problem of bacterial resistance to antibiotics, such as the examples seen in cephalosporins and fluoroquinolones, resulting in the development of the methicillin-resistant Staphylococcus aureus resistant to a full range of antibiotics [11]. It is estimated that by 2050, there could be up to 10 million annual deaths caused by infections that could have been easily cured before resistance if adequate global measures are not taken [12].

One of the complementary therapies in endodontics described as a photo-disinfection technique is photodynamic therapy (PDT); specifically, antimicrobial photodynamic therapy (aPDT). It has other wide therapeutic indications in dentistry, mostly in periodontitis and peri-implant diseases, as well as in the treatment of some oral lesions (lichen planus, candidiasis, lesions induced by human herpes viruses such as recurrent herpes labialis). PDT may also have an indication in the treatment of alveolar osteitis and post-extraction pain [13].

The aPDT mechanism is based on the interaction between a specific wavelength light source and a photosensitizer (PS) (nontoxic natural or synthetic chemical solution, usually a dye applied locally to the target tissues) in the presence of oxygen to induce specific cell damage. There are two ways the PS can react with the biomolecule. The type I reaction, or electron transfer reaction, results in the formation of highly reactive free radicals by removing the electron/hydrogen molecule from the substrate, which then react with the endogenous oxygen molecule, producing reactive oxygen species (ROS) such as hydrogen peroxide, hydroxyl radicals, and superoxide. This reaction causes irreversible cell damage [14,15]. As for the type II reaction, the PS reaches a triple excited state, reacts with the oxygen molecules present at the target cells such as Gram-positive and Gram-negative bacteria, and produces a highly reactive oxygen or singlet oxygen (^1^O_2_) molecule, which leads to microbial cell death by oxidative damage. This affects their plasma membrane, including proteins, lipids, and DNA, without affecting host cell viability [16]. The reactive oxygen singlets have a very short half-life (<0.04 μs) and action ratio (0.02 μm), making their effect very brief and localized. The type II reaction is the most used for combating infections (Figure 1) [17]. This entire process described with aPDT eliminates bacteria in a planktonic state and in biofilms, acting on extracellular biofilm molecules (polysaccharides in the extracellular matrix) by the singlet oxygen released from a photodynamic reaction.

The wavelength used to irradiate the PS is chosen based on the coefficient of absorption of the PS. The corresponding wavelengths described in the literature are 620–700 nm with toluidine blue and methylene blue, 600–805 nm with cyanine, 620–650 nm with hematoporphyrins, 300–500 nm (ultraviolet/blue light) with curcumin and its derivatives, and 660–700 nm with the hytalocyanines [13]. 

The aim of this review is to provide a summary of what we know until now about PDT in relation to endodontics, where the use of antibiotics is a common practice, and if PDT can be an efficient therapy to improve clinical outcomes, reduce complications, and, subsequently, limit antibiotics prescription.

## 2. Materials and Methods 

A literature search was performed in the Web of Science (WOS) and PubMed databases in 2 steps. In the first search, the following terms were used in WOS: ALL = (photodynamic therapy or photochemotherapy or PDT or aPDT) AND ALL = (endodontics or “root canal therapy” or “root canal treatment”) and for PubMed: (“Endodontics” (MeSH)) AND “Photochemotherapy” (MeSH).

The second step was carried out in WOS with the search criteria ALL = (photodynamic therapy or photochemotherapy or PDT or aPDT) AND ALL = (antibiotic resistance) AND ALL = (dentistry) and in PubMed with the MeSH terms (“Photochemotherapy” (MeSH)) AND “Drug Resistance, Microbial” (MeSH).

We included studies that were published from the year 2010 to 2021, in the English language, in vitro and ex vivo studies that compared the effect of PDT in endodontics to other techniques, clinical studies in permanent dentition with mature apices evaluating the disinfection effect of PDT in endodontics, and review articles. Animal studies were excluded and not considered for this review.

## 3. Results

A total of 51 articles were included for evaluation: 27 laboratory studies, 14 reviews, and 10 clinical studies. Data extraction from the included clinical studies is summarized in Table 1, specifying the main author, year of publication, study design, sample size (n), study groups, endodontic pathology to be treated, method of outcome evaluation, follow-up, and clinical outcome reported.

## 4. Discussion

The treatment of acute endodontic infection depends on local microbial reduction by CMD of the root canal space and drainage of the periapical tissue exudate. This local treatment should be sufficient in the absence of additional complications; however, dental practitioners have shown a tendency to overprescribe systemic antibiotics when unnecessary [5]. Some authors surveyed antibiotic prescription habits in the endodontic field around different regions. Silva et al. [18], in a cross-sectional study in Portugal, reported its use by 16–44% of dentists in cases of irreversible pulpitis, 15.8–41.1% in necrotic pulps, and 45% in cases with chronic sinus tract related to the infected tooth. 

In Belgium, Mainjot et al. [19], described antibiotic prescription in the absence of fever by 92.2% of dental professionals, and 54.2% did not perform any local dental treatment after prescription. Members of the Spanish Endodontic Society (AEDE) responded in a survey that they administered antibiotics in 40% of irreversible pulpitis cases and in 53% of necrotic pulps and acute apical periodontitis without swelling [20]. Dormoy et al. [21] surveyed French dentists for the first time to study nonclinical factors that influence antibiotics prescription. Although dentists were aware of the antibiotic resistance public health problem, sometimes they were guided by nonobjective clinical criteria to reassure themselves or their patients. These examples show the lack of adherence to the scientific evidence during the antibiotic prescription protocols in endodontics, which collaborate in the development of antibiotic resistance. 

Bacterial resistance is based on gene changes, and some authors have associated this drug resistance problem with persistent endodontic infections caused by species such as *Prevotella* spp., which is B-lactamase positive, or *E. faecalis* [22,23,24]. In their clinical study, Jungermann et al. [25], identified selected antibiotic bacteria resistance genes performing a polymerase chain reaction of teeth samples with primary and persistent endodontic infection before and after contemporary chemomechanical preparation and medication with calcium hydroxide inside the root canals. They observed the prevalence of beta-lactam resistance genes (blaTEM-1) in primary endodontic infections that was significantly reduced after treatment, whereas tetracycline resistance genes (tetM) found in both primary and persistent endodontic infections were resistant to the endodontic treatment.

To avoid this problem, the use of local application techniques to eliminate residual bacteria from the root canal space have been described, such as the use of local triple antibiotic paste combining ciprofloxacin, metronidazole, and minocycline that can efficiently reduce bacteria but with the risk of developing antimicrobial resistance [26], aPDT has also been investigated in the endodontic field for its promising antimicrobial action. In an in vitro study, Camacho-Alonso et al. [27] compared the antimicrobial effect of triple antibiotic paste, 2% chlorhexidine, ozone therapy, and aPDT and showed significant bacterial reduction in all of them compared with the control group.

Focusing on aPDT, its correct clinical application depends on many factors, such as the pre-irradiation time (time elapsed between the dye application and the beginning of photoactivation), light source power density and duration, PS concentration, wettability to the root canal walls, and oxygen presence at the target cells.

The most commonly used PS in endodontic studies is phenothiazine salts such as toluidine blue (TBO) or methylene blue (MB) because of their amphiphilic nature (both hydrophilic and hydrophobic), which facilitates the staining of both Gram-negative and Gram-positive bacteria responsible for most endodontic infections [28]. PS concentrations described in the literature range from 6.25 to 25 µg/mL for MB and from 10 to 100 µg/mL for TBO [29]. After root canal preparation, PS should be delivered inside the root canal and left in place for 60 s to give time for bacterial staining, and then irradiated for 30 s. Pourhajibagher et al. [30] applied this protocol to eliminate high concentrations of bacteria, and Kosareih et al. [31] reported better dentinal tubule penetration by the PS when the root canal was previously irrigated with 17% EDTA for 2 min.

The corresponding wavelengths activating the light sources for these PS are 630 nm for TBO and 660 nm for MB [32]. Regarding the light source application, Nunes et al. [33] showed that the application is equally effective whether or not an optical fiber is inserted inside the root canal during the photoactivation process.

In relation to power settings, the literature shows a range of heterogeneous parameters between 40 mW and 100 mW with an exposure time from 60 to 240 s [34,35,36].

The success of endodontic therapy can reach 94% when a negative culture is obtained from the root canal prior to obturation [37], so one of the main challenges in endodontic infections is combating the presence of multispecies bacteria attached to the surfaced of the root canal, forming a biofilm. 

This biofilm disruption inhibits the transfer of antibiotic resistance genes among bacteria and interferes with bacterial colonization [17]. Additionally, up to now, we did not find any report of bacterial resistance to aPDT. This can be attributed to its nonspecific mechanism of action by the oxygen singlet, which reacts against several cell components, unlike the key-lock principle of antibiotics in which bacterial mutation can modify their susceptible receptors [29]. Nevertheless, some authors [38,39] attempted to develop aPDT resistance by repeated sub-lethal doses and the regrowth of microorganisms and by repeated cycles of partial inactivation, but with no success.

In an in vitro study, López Jiménez et al. [40] compared the effect of different dyes (TBO and MB) and light sources (diode laser 670 nm and LED 628 nm) alone or in combination on biofilms with Enterococcus faecalis. The use of light therapy alone or in combination with the dyes altered the biofilm topography, such as bacterial wall destruction, loss of cell morphology, or leakage of the intracellular contents, whereas the dyes alone did not induce morphological changes. Additionally, bacterial surface roughness increased after treatment by the same combination therapy groups (TBO with 628 nm and MB with 670 nm).

Some authors optimized the PS biofilm penetration capacity and increased its affinity to bacterial cell membrane by combining it with rose bengal-functionalized chitosan nanoparticles [41,42]. Afkhami et al. [43] showed better biofilm penetration using a diode laser with silver nanoparticles and indocyanine green.

In their review, Cieplik et al. [44] stated the benefits of aPDT to deactivate biofilms and highlighted its promising effects. However, most of the laboratory studies used different experimental biofilm models with different culture protocols, resulting in heterogeneous studies difficult to compare.

Ex vivo laboratory studies tried to reproduce the clinical scenario of a multispecies biofilm using necrotic pulps. Ng et al. [45] compared microbiological samples from necrotic root canals of recently extracted human teeth before and after treatment with CMD alone or in combination with aPDT and reported significant bacterial reduction after aPDT application, with lower bacterial concentrations in dentinal tubules at a depth up to 485 μm. 

The superior aPDT results were also confirmed by Hoedke et al. [46] in another ex vivo experiment detecting more bacterial reduction belonging to a planktonic and adherent bacterial multispecies biofilm in root canals after CMD combined with 2% chlorhexidine and aPDT, even after 5 days of further incubation.

To date, there have been few recent clinical trials on aPDT in relation to the endodontic field, although those that we reviewed show a promising tendency to obtain better results with the aPDT approach in addition to the conventional CMD (Table 1). 

In a quasi-controlled study on endodontically treated teeth with persistent infection, Garcez et al. [47] took three bacterial culture samples from each tooth, the first one after accessing the root canal space and comparing it with another culture after CMD and with a final one following aPDT. 

The first microbiological sample confirmed the presence of at least one microorganism resistant to antibiotics of *Enterococcus* sp., *Prevotella* sp., *Actinomyces* sp., *Peptostreptococcus* sp., *Streptococcus* sp., *Fusobacterium* sp., *Porphyromonas* sp., *Enterobacter* sp., and *Propionibacterium* sp., and with different degrees of antimicrobial resistance to erythromycin, ampicillin, penicillin G, vancomycin, cephalosporin, clindamycin, chloramphenicol, and tetracycline. The aPDT completely eliminated the microbial load of all 30 teeth, proving the efficacy of this approach toward drug-resistant bacteria found in persistent infections.

Zorita-García et al. [48] in a microbiological clinical study in forty-two single rooted necrotic posterior teeth obtained three samples from the same tooth. In the initial sample taken after accessing the root canal, a mean value of 113.5 ± 130 colony forming units (CFU) per tooth was detected. After CMD, the mean CFU per tooth decreased to 26.52 ± 72, which was even more reduced in the third sample after aPDT to 4.2 ± 13 CFU/tooth.

aPDT application in endodontic periapical surgery was also evaluated by the same author in a clinical study [49]. After cleaning the surgical area and preparing the root end for sealing, a microbiological culture was taken to compare it to another culture obtained after aPDT application. A mean bacterial reduction of 1.5 log was reported, indicating its additional benefits over conventional surgery alone. The author reported periapical healing of 78% in a 3-year follow-up period. However, the culture-based analysis has low sensitivity and can fail to detect culture-difficult, or yet, uncultured bacteria.

Based on this study, Vieira et al. [50] improved the microbiological detection method using a quantitative real-time polymerase chain reaction (qPCR) in a case series study on 16 patients with 19 teeth programmed for periapical surgery. They found a significant reduction in total bacterial and streptococci levels after applying aPDT and reported high healing rates with a mean follow-up of 16 months.

Abu Hasna et al. [51] used aPDT as an adjunctive disinfection technique to manage a failing symptomatic endodontically treated tooth due to root perforation near the apex. They used aPDT during endodontic retreatment to remove all residual bacteria from the root canal space before obturation, as well as during the apicoectomy surgery on the same tooth to disinfect the surgical area before root end sealing. The apical area was obturated with bioceramic cement, and the bone defect was bone grafted. A twelve-month follow-up using 3D cone beam computed tomography in the treated area showed complete healing with neobone formation, indicating a successful outcome.

Moreira et al. [52] reported, in two cases of failed endodontic retreatments, the possibility of applying seven to ten sessions of aPDT via the sinus tract to avoid surgical intervention and antibiotic prescription, where both cases showed the complete healing of the sinus tract with a periapical bone repair and they were asymptomatic.

Efficient microbial eradication reduces healing time and offers better clinical outcomes, as shown in a randomized controlled clinical trial by De Miranda et al. [53]. They treated necrotic teeth with aPDT in the experimental group, where they observed better healing and lower periapical index (PAI) scores at the 6-month follow-up. Nevertheless, longer follow-up periods are recommended to reach a more reliable conclusion. 

Conejero et al. [54] in a retrospective clinical study analyzed two groups. The group where aPDT was applied as an adjunctive therapy showed a shorter periapical healing time (15 ± 9.33 months) compared to conventional CMD alone (20.35 ± 22.1 months).

The superior bacterial elimination is in accordance with the randomized controlled clinical trial by Rabello et al. [55] that evaluated the antimicrobial effect of aPDT in one-visit versus two-visit cases, with calcium hydroxide intracanal medication between appointments. They described the existence of a significant bacterial reduction in one-visit cases with aPDT but no further benefits in the two-visit approach. The effectiveness of a single-visit approach using aPDT was confirmed by the randomized controlled clinical trial by Asnaashari et al. [56] in endodontic retreatment cases, concluding even a superior microbiological eradication after applying aPDT in one session versus a calcium hydroxide dressing in two sessions. The ability to perform endodontic treatment in a single visit permits an immediate coronal restoration, which reduces the possible bacterial contamination from the oral flora during the 2-week waiting period of the two-visit approach.

## 5. Conclusions

Our review projects the bacterial resistance problem focusing on the endodontic field and the challenges of persistent infections with antimicrobial-resistant strains. Antibiotic prescription should be conducted with caution after an adequate evaluation of the clinical situation to avoid overprescription.

Local aPDT treatment showed a promising tendency toward significant bacterial elimination, which has also proven effective in biofilms disruption. This adjunctive therapy can improve the clinical outcome, increasing the success rate in conventional and surgical endodontic procedures due to its efficient elimination of resistant bacteria. Therefore, it could avoid the appearance of endodontic complications that involve the use of a systemic antibiotic prescription. In addition, it could be an alternative to local intracanal antibiotic therapy, avoiding possible antibiotic resistance.

This therapy seems to be safe and predictable, and the literature reported no resistance toward it to date. More high-evidence clinical trials are needed with long-term follow-up to confirm the superiority of aPDT as an adjunctive tool and spread its use among dentists.

## Figures and Tables

**Figure 1 antibiotics-10-01106-f001:**
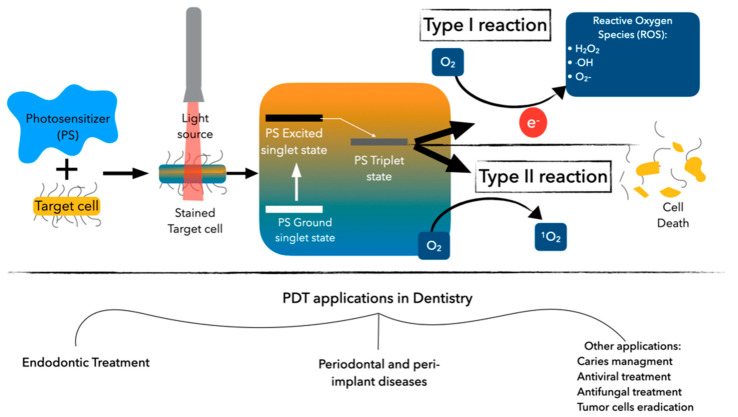
Photodynamic therapy process and its application in dentistry.

**Table 1 antibiotics-10-01106-t001:** Data extraction of the included clinical studies.

Author	Year	Study Design	Sample Size (n)	Study Groups	Endodontic Pathology	PS and Concentration	Pre-Irradiation Time	Light Source and Wavelength	Power	Power Density	Irradiation Time	Outcome Evaluation	Follow-Up	Clinical Outcomes
Miranda et al.	2017	Randomized controlled clinical trial	32 patients (16 in each group)	Control group (CMD + calcium hydroxide), PDT group (CMD+PDT+Cal hydroxide)	Primary endodontic infections	MB (25 μg/mL)	5 min	Diode laser (660 nm)	100 mW	Not reported	5 min	PAI	Baseline, 3 and 6 months	Statistically significant improvement Periapical Index Score at 6 months
Rabello et al.	2017	Case series	24 patients (12 in each group)	1 visit PDT and 2 visits calcium hydroxide and PDT	Primary endodontic infection	MB (0.1 mg/mL)	60 s	Diode laser (660 nm)	60 mW	Not reported	2 min	Microbiological endodoxins analysis	No follow up was reported	PDT significant reduction of 1 visit compared to control group but not significant in 2 visits with calcium hydroxide
Asnaashari et al.	2016	Case series	20 patients	2 sessions with calcium hydroxide intracanal dressing or a single visit with adjunctive aPDT	Persistent endodontic infection	TBO (0.1 mg/mL)	5 min	LED (620–640 nm)	Not reported	2–4 mW/cm^2^	60 s	Microbial reduction by culture samples	No follow up was reported	Decrease in number of colonies was more evident in aPDT group
Garcez et al.	2010	Quasi-controlled clinical trial	30 patients	No groups. 3 microbiological culture samples were taken in the same tooth of each patient	Persistent endodontic infection	Conjugate between polyethylenimine (PEI) and chlorin (e6) (60 μmol/L)	2 min	Diode laser (660 nm)	40 mW	Not reported	4 min	Microbial reduction by culture samples	No follow up was reported	10 root canals after CMD (eliminated microorganism, while all 30 had complete elimination after PDT)
Garcez et al.	2015	Case series	28 teeth (from 22 patients)	Endodontic surgery (apicoectomy) with PDT	Persistent endodontic infection	MB (60 μM)	3 min	Diode laser (660 nm)	40 mW	Not reported	3 min	-Microbial reduction by culture samples. -Radiographic follow up.	36 months radiographic follow up	-Significant reduction in bacterial culture samples after aPDT application -average periapical radiographic lesion reduction of 78%
Vieira et al.	2018	Randomized controlled clinical trial	19 teeth treated in 16 patients	Endodontic surgery (apicoectomy) with PDT	Persistent endodontic infection	MB (0.01%)	Not reported	Diode laser (660 nm)	40 mW	Not reported	3 min	Evaluate healing: (complete, incomplete, uncertain, unsatisfactory) Radiographic and clinical evaluation: rigid and loose	12–21 months (mean of 16 months)	Statistically significant bacterial reduction after PDT and 93% success rate (loose criteria) and 73% success using rigid criteria
Abu Hasna et al.	2020	Case report	1 patient with endodontic retreatment and apicoectomy using adjunctive aPDT	-	Persistent endodontic infection	MB (0.005%)	5 min	Diode laser (660 nm)	Not reported	100 mW/cm^2^	2 min	Clinical and radiographic outcome	30 days and 12 months	Twelve-month cone beam computed tomography follow-up showed bone neoformation at the periapical area indicating success of the treatment
Zorita García et al.	2019	Quasi-controlled clinical trial	42 posterior single rooted teeth (33 patients)	No groups. 3 microbiological culture samples were taken in the same tooth of each patient	Primary endodontic infection	TBO (concentration was not reported)	2 min	LED (630 ± 20 nm)	Not reported	2000 mW/cm^2^	2 cycles of 30 s each	Microbial reduction by culture samples	No follow up was reported	Significant reduction in CFU/tooth was achieved after aPDT application in all cases.
Moreira et al.	2015	Case reports	2 cases with failed endodontic retreatments and with persistent sinus tract.	-	Persistent endodontic infection	MB (0.01%)	4 min	AsGaAl diode laser (660 nm)	40 mW	Not reported	63 s	-Clinical sinus tract healing. -CBCT.	2 and 4 years of follow-up, respectively	Healing of the sinus tract was achieved with periapical bone repair confirmed by CBCT after seven to ten sessions via sinus tract.
Conejero et al.	2021	Retrospective study	100 teeth treated with conventional CMD and 114 teeth received adjunctive aPDT	CMD or CMD + aPDT	Primary and persistent infections	TBO (0.1 mg/mL)	Not reported	LED (360 nm)	Not reported	2000 mW/cm^2^	30 s	-Radiographic PAI index score. -Clinical signs and symptoms.	No follow up (retrospective study)	-Success rate for CMD group was 94.7% and for CMD + aPDT was 97.2%. -Periapical healing in CMD group 20.35 ± 22.1 months, CMD + aPDT group was 15 ± 9.33 months.

CMD (chemo-mechanical disinfection), PDT (photodynamic therapy), MB (methylene blue), PAI (periapical score index), aPDT (antimicrobial photodynamic therapy), TBO (toluidine blue O), LED (light-emitting diode), AsGaAL (Arsenide Gallium Aluminum).

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
