# Peer review of "Photodynamic Therapy in Endodontics: A Helpful Tool to Combat Antibiotic Resistance? A Literature Review"

_antibiotics, 2021, doi:10.3390/antibiotics10091106_

Round 1
Reviewer 1 Report
- As a review, illustrations are needed to enhance the readability of the article.
- First look at he tittle"Photodynamic therapy: Is the future solution to antibiotic resistance in the dental practice? A literature review", I will think the PDT is alternative methods for fighting dental infection. But from the abstract they just summarize the adjunctive PDT therapy cases , which seem unreasonable.
- If photodynamic as a single therapy method also concludes in this review, for example, reference 58. I think this manuscript is not comprehensive enough in my opinionin
Author Response
Reviewer 1
1-Considering your suggestions, we revised our search strategy and on reviewing the literature in photodynamic therapy (PDT), we found it more feasible to focus our review on the endodontics field as it is less investigated compared to periodontics and peri-implantitis. Also we expanded our search strategy including web of science (WOS) database where more relevant studies were retrieved and included. The link between the benefits of the PDT and its possible role in mitigating the antibiotic resistance problem was argued throughout the manuscript, for example we projected the equivalent efficacy of PDT to the local triple antibiotic paste which can cause bacterial resistance .
2-The manuscript was sent for a professional English revision.
3-The abbreviations were revised indicating its meanings throughout the text.
4-The introduction section was changed and reorganized to be focused on the scope of our review in the endodontics field.
5-Web of science database was considered and more articles were retrieved and included in our review.
6-MeSH terms were revised and corrected.
7-With the new search strategy more studies were included related to the topic.
8-inclusion criteria were defined.
9-We specified the search date (articles were included from 2010 till 2021).
10-A link between the benefits of the photodynamic therapy as an adjunctive method with antibiotic resistance was highlighted.
11-Literature revision was done to make we included most relevant studies.
Reviewer 2 Report
Dear authors, the topic is very hot due to the antibiotic resistence problem and the article is very interesting. I got this is only a literature review not a systematic one, but you have to try to be more inclusive and to make clearer and larger the results paragraph. Please add in the results the same paragraph you proposed in the discussion. In addition in the discussion you must amplify and clarify when you talk about the limitation of your study and possible future investigation that are needed. Moreover in the last part of the discussion you have to introduce other possible solution that have been proposed to reduce the antibiotic use such as the ozone therapy, using the oxidative stress too or lactoferrin-based solutions to have an antioxidant effect on soft tisse and an oxidative effect on bacteria.
Regarding these two topic that I have introduced I suggest to cite these two following recent articles, but you have to introduce also other possibile similar solution and cite other article in the discussion:
Cosola S, Giammarinaro E, Genovesi AM, Pisante R, Poli G, Covani U, Marconcini S. A short-term study of the effects of ozone irrigation in an orthodontic population with fixed appliances. Eur J Paediatr Dent. 2019 Mar;20(1):15-18. doi: 10.23804/ejpd.2019.20.01.03.
Marconcini, S.; Giammarinaro, E.; Cosola, S.; Oldoini, G.; Genovesi, A.; Covani, U. Effects of Non-Surgical Periodontal Treatment on Reactive Oxygen Metabolites and Glycemic Control in Diabetic Patients with Chronic Periodontitis. Antioxidants 2021, 10, 1056. https://doi.org/10.3390/antiox10071056
Author Response
Reviewer 2:
we revised our search strategy and on reviewing the literature in photodynamic therapy (PDT), we found it more feasible to focus our review on the endodontics field as it is less investigated compared to periodontics and peri-implantitis. Also we expanded our search strategy including web of science (WOS) database where more relevant studies were retrieved and included. The link between the benefits of the PDT and its possible role in mitigating the antibiotic resistance problem was argued throughout the manuscript, for example we projected the equivalent efficacy of PDT to the local triple antibiotic paste which can cause bacterial resistance .
2-A summary table for the clinical studies included was added, including study design, study groups and outcomes.
3-The mechanism of action of the photodynamic therapy was represented by a figure from our design.
4-We reconsidered a more critical revision and restructure of the manuscript and we focused it on one topic and its link to the antibiotic resistance problem.
Reviewer 3 Report
The authors present a review focusing on the use of photodynamic therapy in dentistry and if it can be the solution to antibiotic resistance. This is an interesting topic; however, the review methodology and included articles raise some doubts.
The English language needs revision through the text.
Line 42: the first time abbreviations are used, please indicate their meaning.
The introduction section provides the necessary information but should be improved for clarity. For example, the change of topic is abrupt, and the text lacks a clear organization.
Materials and methods: why choose Pubmed, Medline, and Scielo? First, PubMed and Medline present very close results. Second, why not include other databases as Embase or Web of Science, for instance?
Line 117: “antimicrobial photodynamic therapy” is not a MeSH term. Please correct it.
The search strategy is incomplete, which limits the obtained results. A more complete search formula should be used, which will allow obtaining more results.
The authors used a somehow systematic review methodology to perform the present review. In lines 119-120, the authors refer that “studies were identified with potential for inclusion”. What criteria were applied to define the studies to be included and excluded?
Please provide the date for the last search.
The results/discussion section describes the results obtained in the different studies, but in most cases, they refer to bacteria elimination and fail to report or discuss antibiotic resistance. Thus, the link between the two topics is missing, which limits the review interest.
Since the search strategy was incomplete, relevant studies are missing from the results.
Although being an interesting topic, I find this review to present limited information and interest. Therefore, I suggest the authors include other databases and improve the search formulas. Also, the results/discussion section will benefit more studies to be included and their information detailed, besides a more precise correlation with the topic of antibiotic resistance.
Author Response
Reviewer 3
we revised our search strategy and on reviewing the literature in photodynamic therapy (PDT), we found it more feasible to focus our review on the endodontics field as it is less investigated compared to periodontics and peri-implantitis. Also we expanded our search strategy including web of science (WOS) database where more relevant studies were retrieved and included. The link between the benefits of the PDT and its possible role in mitigating the antibiotic resistance problem was argued throughout the manuscript, for example we projected the equivalent efficacy of PDT to the local triple antibiotic paste which can cause bacterial resistance .
-Ozone therapy and lactoferin based solutions are other alternatives for bacterial elimination, but our review is focused on the photodynamic therapy.
Reviewer 4 Report
This paper is to provide a summary about the photodynamic therapy in relation with endodontics, periodontal and peri-implant fields. However, the paper is not well structured and presented. Throughout the paper, there is no any figures. I suggest a major revision.
- I require the structure of the paper to be revised. Starting with a graphical abstract showing the relation between the photodynamic therapy and the dental field. From that figure, the paper is designed straightforward
- A summary table of the related papers in the field must be included. This helps the reader easier to follow and catch important information
- The mechanisms presented in this paper should be explained by figures.
- The review paper need much effort and presentation. Please include figures
Author Response
Reviewer 4:
we revised our search strategy and on reviewing the literature in photodynamic therapy (PDT), we found it more feasible to focus our review on the endodontics field as it is less investigated compared to periodontics and peri-implantitis. Also we expanded our search strategy including web of science (WOS) database where more relevant studies were retrieved and included. The link between the benefits of the PDT and its possible role in mitigating the antibiotic resistance problem was argued throughout the manuscript, for example we projected the equivalent efficacy of PDT to the local triple antibiotic paste which can cause bacterial resistance .
2-We reconsider our terms and argument for the advantages of the photodynamic therapy(PDT) and the link between the antibiotic resistance problem and not referring to PDT as an alternative to antibiotics.
3-We focused our scope on the application of PDT in endodontics, to be more organized and a more comprehensive manuscript.
Round 2
Reviewer 1 Report
After revison, the manuscript can be accepted for publication in present form
Author Response
Dear reviewer 1,
Thank you again for your time revising our modified manuscript. We are glad to hear your approval to the editor for publication.
Regards.

Reviewer 3 Report
The authors present a review focusing on the use of photodynamic therapy in endodontics and if it can be the solution to antibiotic resistance. The present version is significantly different from the manuscript that was initially submitted. It now focuses only on endodontics which I agree is a good option due to PDT application and potentiality in this dental field. Nevertheless, I still have some considerations.
Lines 13-14: overprescription is not the only cause for antibiotic resistance. Please correct the sentence.
Why was the literature search limited to articles published from 2010?
Line 20: photodynamic therapy is not a MeSH term.
Line 35: please remove “the photodynamic therapy (PDT)”.
Lines 35-38: the variety of bacterial strains is not the only cause for the complexity of root canal treatment. Several other causes as teeth anatomy, teeth location… also contribute. Please correct the sentence.
Line 89: figure 1 is missing.
The search strategy is incomplete, which limits the obtained results. Therefore, a more complete search formula should be used, which will allow getting more results. For instance, for WOS, the formula should include ALL=(photodynamic therapy Or photochemotherapy Or PDT or aPDT) AND ALL=(endodontic* Or “root canal therapy” Or “root canal treatment”). The same applies to the other search formulas.
Did the authors also considered in vivo (animal studies)?
Lines 177-180: I suggest this information be moved to the introduction section.
I suggest table 1 be added in the manuscript file and not as a supplementary table since it synthesizes the clinical studies results.
Since this review is focused on PDT, I suggest adding the PDT parameters for each study in the table (photosensitizer used, drug-light interval, wavelength…)
Why did the authors not make similar tables to ex vivo and in vitro studies?
Table 1: please place the table title above the table and add a table caption describing the meaning of the abbreviations.
Author Response
Dear reviewer 3,
Thank you again for your time revising our modified manuscript.
The following points have been considered:
1-Lines 13-14: we corrected the sentence in order not to refer to the overprescription as the only cause of antibiotic resistance.
2-Our Literature search was limited to articles from the year 2010 till 2021 as an inclusion criteria to provide a recent update about aPDT in endodontics and its link to mitigate antibiotic resistance problems, also previous reviews included studies before 2010 as the following examples:
-Plotino G, Grande NM, Mercade M. Photodynamic therapy in endodontics. Int Endod J. 2019 Jun;52(6):760-774.
-Singh S, Nagpal R, Manuja N, Tyagi SP. Photodynamic therapy: An adjunct to conventional root canal disinfection strategies. Aust Endod J. 2015 Aug;41(2):54-71.
3-Line 20: We corrected the MeSH term from photodynamic therapy to photochemotherapy.
4-Line 35: We removed ¨the photodynamic therapy (PDT)¨
5-Lines 35-38: We modified the sentence adding other contributing factors that presents challenges for successful rooth canal therapy among them the variety of the bacterial strains in endodontic infections.
6-Line 89: We included Figure 1 in the manuscript
7-The search strategy was modified and corrected with your indications and three more clinical studies were included.
8-We specified in materials and methods section that animal studies were excluded as we focused our scope on clinical application and outcomes andlaboratory studies using human teeth.
9-Lines 177-180: This information was considered to be moved to the introduction section.
10-We added table 1 to the manuscript file and not as a supplementary data.
11-The table we added synthesizes the treatment parameters and the clinical outcomes of this adjunctive therapy as the interest of this review to provide a practical data and guide for future clinical trials to help filling the gaps in the literature on a clinical level.
12-We edited the table title to be placed above the table and we added a table caption describing the meaning of the abbreviations.

Reviewer 4 Report
In my opinion, this review have not reach the quality of the SCI journal like Antibiotics; although, the authors try to improve the quality of the manuscript after revision. My comments were also not seriously considered by the authors. The authors should put more efforts to improve their manuscript.
Author Response
Dear reviewer 4,
Thank you again for your time revising our modified manuscript.
Considering your previous comments in round 1 for major revisions, we modified our manuscript and reduced the scope of the review to increase its quality in the time frame given by the editor.
Regarding your previous comments they were all considered as follows:
1-We elaborated an illustrative figure summarizing the PDT applications in dentistry and explaining the mechanism of action of the PDT and was reffered to in the text as Figure 1. However, we think there was a technical error when we uploaded the corrected version and the figure was not attached. We re-inserted the figure in the text.
2-The whole structure of the paper was revised, reorganized and was also sent for professional English correction.
3-The manuscript was restructured in a way to link the benefits of the aPDT in endodontics and its ability to combat antibiotic resistant microorganisms, as well as its efficacy in persistent infections which can avoid the need for future antibiotic prescription.
4-A summary table of the clinical studies included was inserted as a supplementary data. Nevertheless, we modified it to be inserted in the main paper as it synthesizes the results in our review.
Best regards.
